# Simple Calibration of Three-Axis Accelerometers with Freely Rotating Vertical Bench

**DOI:** 10.3390/s25133998

**Published:** 2025-06-26

**Authors:** Federico Pedersini

**Affiliations:** Department of Computer Science, University of Milan, Via Giovanni Celoria 18, I-20133 Milan, Italy; federico.pedersini@unimi.it

**Keywords:** accelerometers, dynamic calibration, IMU, MEMS, modeling, sensor

## Abstract

This article presents a dynamic calibration procedure for triaxial accelerometers characterized by a very simple and low-cost setup, where the calibration bench consists of a vertically, freely rotating wheel. To keep the setup as simple as possible, the necessary prior knowledge about the geometry and the motion of the bench has been minimized: the only required constraint is the verticality of the rotation plane, which can be simply achieved in practice by means of a level tool. No prior knowledge is required about the bench rotation, as the calibration procedure estimates both the accelerometer parameters and the bench motion. The precision achieved by the proposed calibration has been tested on synthetic data to prove the absence of estimation biases and to evaluate the potential accuracy, and on real data (from a MEMS accelerometer) to evaluate the achievable precision.

## 1. Introduction

Accurate acceleration measurements are of key importance in all motion analysis applications using IMUs, since velocity and displacement are obtained by integrating acceleration. Therefore all systematic errors in the acceleration data (like offsets and gain disparities) cause drifts that severely affect the motion estimates [1].

Many approaches to calibration of accelerometers have been proposed in the literature [2,3,4,5,6,7,8,9,10,11,12]. All these techniques can be grouped in two main categories: static and dynamic calibrations. In static methods, acceleration is sampled while the sensor remains stationary and oriented in various different attitudes [2,4,7,8,9,10], whereas in dynamic contexts acceleration is recorded as the sensor undergoes a controlled movement [3,6,11,12,13,14,15]. The dynamic methods have the advantage of needing a single data capture session, which makes the data acquisition process simpler with respect to static methods, typically needing to carry out several acquisitions, one for each attitude. The calibration accuracy of dynamic methods strongly depends on the extent and precision of prior knowledge about the motion applied to the sensor. Methods that utilize high-precision mechanical setups generally achieve the highest levels of accuracy, as they depend on precise knowledge of the sensor’s movement [6,9,11,13]. However, such equipment is usually expensive and complex, which is why these methods are generally reserved for high-grade accelerometers, such as those used in seismic applications [6].

In contrast, for applications using commercial-grade MEMS sensors, where the cost of such calibration equipment is typically prohibitive, it is more reasonable to adopt the opposite approach: a procedure based on a maximally simple mechanical setup, providing minimal prior knowledge about the sensor motion, where all the unknown motion information is determined together with the accelerometer parameters. To some extent, such approaches [4,10] can be considered a self-calibration, as they estimate also the setup motion together with the desired sensor model. In this work, I pursue such an approach by developing a calibration method for commercial-grade triaxial accelerometers that leverages a particularly simple and low-cost mechanical setup, constructed from common materials and without requiring precise mechanical constraints.

This method can be considered an evolution of the calibration procedure in [16] based on the following aspects:Owing to the use of a vertical rotation plane, the calibration problem is fully determined in the present approach. In contrast, the method in [16] involves an underdetermined problem that requires the user to perform a manual measurement prior to calibration. In this work, no preliminary measurements are necessary; all parameters are identified directly through the calibration process.The vertical rotation plane also allows us to address the problem of the optimal orientation with which the sensor should be positioned on the calibration bench. The analysis (in Section 3.2) leads to the proposal of a ‘good orientation’ that guarantees a valid and steady signal/noise ratio along all three axes.

The results presented in Section 5 demonstrate that this approach significantly improves the achievable accuracy of the estimated parameters compared to the method in [16].

The article is organized as follows: Section 2 and Section 3 describe the calibration bench and the consequent dynamic acceleration model applied to the sensor under test; Section 4 describes the calibration algorithm; and Section 5 reports the performance yielded by the algorithm with both synthetically generated data and experimental data acquired from a three-axis MEMS accelerometer.

## 2. The Vertical Bench Setup

A schematic of the proposed calibration setup is shown in Figure 1. The calibration bench features a plate mounted on a freely rotating wheel. The wheel is mounted with a horizontal axis of rotation, allowing the bench to rotate within a vertical plane.

To perform a calibration, the accelerometer under test is mounted on the plate, with a counterbalance applied on the opposite side. The bench is then manually spun and, while it freely rotates, the acceleration measured by the sensor under test is sampled for several turns of the bench. These data are the calibration input.

The only imposed mechanical constraint in the construction of the bench is the verticality of the rotation plane. This mechanical constraint can be simply achieved adjusting the inclination of the bench by means of a level tool. I show in Section 2.2 that this precision is sufficient to consider the bench vertical for the calibration accuracy. All other geometric and dynamic parameters—such as the sensor’s orientation on the plate, its rotation radius, and the time-varying angular position of the bench—are unknown and will be estimated by the calibration procedure, alongside the accelerometer parameters, which are the primary focus of the calibration.

### 2.1. Basic Acceleration Model

The basic model describing the accelerations applied to the sensor due to the bench rotation is shown in Figure 2. An object undergoing a circular motion in a vertical plane is subject to the following accelerations:Gravity *g*, directed downwards;Centripetal acceleration $r\;\dot{\vartheta }{\left(t\right)}^{2}$, directed toward the center of rotation;Angular (tangential) acceleration $r\;\ddot{\vartheta }\left(t\right)$, tangent to the circular trajectory.

In the above, ϑ(t) denotes the angular position of the sensor on its circular trajectory; therefore $\dot{\vartheta }\left(t\right)$ defines its angular velocity and $\ddot{\vartheta }\left(t\right)$ the angular acceleration.

Referring to Figure 2, let us consider the local reference frame 〈XYZ〉 which is rigidly connected to the bench and whose origin is located on the sensor; the *X* axis corresponds to the radial direction, *Y* to the tangential direction, and *Z* is perpendicular to the plane of rotation. The components of the applied acceleration a(t) along these axes are given by(1)$$a\left(t\right)=\left[\begin{array}{cc} a_{X}\left(t\right)= & r\;{\dot{\vartheta }}^{2}\left(t\right)+g\;\cos\vartheta \left(t\right)\\ a_{Y}\left(t\right)= & r\;\ddot{\vartheta }\left(t\right)\;+g\;\sin\vartheta \left(t\right)\\ a_{Z}\left(t\right)= & 0\; \end{array}\right].$$

### 2.2. Verticality Assumption

As said before, the verticality of the bench is achieved by manually adjusting its inclination by means of a common level tool. The sensitivity of such tools is typically ±0.1% (±1 mm/m), corresponding to a detectable angle of $10^{-3}$ rad $\approx 0.06^{\circ }$.

To assess the impact of any deviation from verticality of the rotating bench on the accuracy of the proposed calibration, we consider the ’nearly vertical’ model shown in Figure 3, where the rotation axis is tilted by an angle δ with respect to the horizontal. In this model, the three sources of acceleration are no longer coplanar: while centripetal and angular accelerations lie in the tilted rotation plane, gravity remains vertical.

As shown in Figure 3, the components of the acceleration along the axes of the local reference frame 〈XYZ〉, due to the tilted rotation plane, turn out now to be the following:(2)$$a^{\prime }\left(t\right)=\left[\begin{array}{cc} a_{X}^{\prime }\left(t\right)= & r\;{\dot{\vartheta }}^{2}\left(t\right)+g\;\cos\delta \cos\vartheta \left(t\right)\\ a_{Y}^{\prime }\left(t\right)= & r\;\ddot{\vartheta }\left(t\right)\;+g\;\cos\delta \sin\vartheta \left(t\right)\\ a_{Z}^{\prime }\left(t\right)= & g\;\sin\delta \end{array}\right]$$

Considering a sensitivity of ±1 mm/m for a common level tool, we can therefore assume an angular tilt $\left|\delta_{max}\right|\leq 10^{-3}\;$ rad. This leads to the following constraints for sinδ and cosδ in (Equation 2): $$\begin{array}{cccc} \; & \left|\sin\delta \right| & < & 10^{-3}\\ \; & \cos\delta =\sqrt{1-\sin^{2}\delta }\;\;≊\;\;1-\frac{\sin^{2}\delta }{2} & > & 1-5\;\cdot \;10^{-7}\;≊1 \end{array}$$
This allows us to conclude the following:cosδ is definitely negligible in $a_{X}^{\prime }\left(t\right)$ and $a_{Y}^{\prime }\left(t\right)$, as its effect would only become observable beyond the sixth decimal place, which is well below the sensitivity of any MEMS accelerometer. Consequently, $a_{X}^{\prime }\left(t\right)≊a_{X}\left(t\right)\;\;\mathrm{and}\;\;a_{Y}^{\prime }\left(t\right)≊a_{Y}\left(t\right)$.sinδ introduces a constant systematic error on $a_{Z}\left(t\right)$, as $\left|a_{Z}^{\prime }\left(t\right)\right|<10^{-3}g≊1$ mg. This error is not so negligible as cosδ, but it can be nevertheless neglected, as its maximum value lies below the noise floor of commercial-grade MEMS accelerometers [17,18].

We can therefore reasonably assume that, for those accelerometers, a manual adjustment of the bench verticality using a common level is accurate enough to consider the proposed acceleration model (Equation 1) valid for this calibration.

## 3. The Dynamic Calibration Model

### 3.1. Modeling the Rotation Speed

Due to frictions and air drag at the moving parts of the bench, its rotation speed $\dot{\vartheta }\left(t\right)$ slowly decreases during the free rotation.

Experimental free-rotation tests conducted in [16] demonstrated that the rotation speed $\dot{\vartheta }\left(t\right)$ closely follows a decreasing cubic polynomial function of time. I therefore model the rotation speed $\dot{\vartheta }\left(t\right)$ and consequently the phase ϑ(t) and the angular acceleration $\ddot{\vartheta }\left(t\right)$ as follows:(3)$$\begin{array}{cc} \; & \dot{\vartheta }\left(t\right)=\omega_{0}+\omega_{1}\;t+\omega_{2}\;t^{2}+\omega_{3}\;t^{3}\\ \to & \vartheta \left(t\right)=\vartheta_{0}+\omega_{0}\;t+\frac{\omega_{1}}{2}\;t^{2}+\frac{\omega_{2}}{3}\;t^{3}+\frac{\omega_{3}}{4}\;t^{4}\\ \to & \ddot{\vartheta }\left(t\right)=\omega_{1}+2\;\omega_{2}\;t+3\;\omega_{3}\;t^{2} \end{array}$$
According to this model, the acceleration a(t) in (Equation 1) is a function of the rotation radius *r* and the phase ϑ(t), which, in turn, is described by the five motion parameters $\langle \vartheta_{0},\omega_{0},\omega_{1},\omega_{2},\omega_{3}\rangle$ in (Equation 3). Combining (Equation 1) and (Equation 3) results in a(t) being modeled by the following six parameters: $\langle r,\vartheta_{0},\omega_{0},\omega_{1},\omega_{2},\omega_{3}\rangle$. These are the bench parameters, which must be estimated through calibration, along with the accelerometer parameters.

### 3.2. Sensor Orientation

Since no geometrical constraint is imposed on how the accelerometer is mounted on the bench, the actual sensor orientation is unknown. As shown in Figure 4, the misalignment between the sensor reference frame $\langle xyz\rangle$ and the bench reference frame $\langle XYZ\rangle$ can be modeled as a 3D rotation. Naming s(t) the acceleration sensed by the accelerometer along its sensing axes $\langle xyz\rangle$, the relationship between s(t) and the actual acceleration a(t) along the axes of the bench frame $\langle XYZ\rangle$ is(4)$$s\left(t\right)=\left[\;\begin{array}{c} s_{x}\\ s_{y}\\ s_{z} \end{array}\;\right]=R\;a\left(t\right)\;,\;\;R=\mathrm{rotmat}\left(\theta ,\phi ,\psi \right)$$
where R is the *misalignment* rotation matrix, defined by its three Euler angles 〈θ,ϕ,ψ〉. Actually, since $a_{Z}\left(t\right)=0$, a 2D acceleration, $a_{XY}$, and a corresponding 3×2 matrix $R_{32}$ could be used in the model, as follows:(5)$$\left[\;\begin{array}{c} s_{x}\\ s_{y}\\ s_{z} \end{array}\;\right]=R_{32}\;a_{XY}\left(t\right)=\left[\;\begin{array}{cc} r_{11} & r_{12}\\ r_{21} & r_{22}\\ r_{31} & r_{32} \end{array}\;\right]\left[\;\begin{array}{c} a_{X}\\ a_{Y} \end{array}\;\right];$$
however, since R and $R_{32}$ are both functions of three Euler angles $\left(\theta ,\phi ,\psi \right)$, there is no advantage in using the reduced model given in (Equation 5), as it does not reduce the number of unknowns in the problem.

#### A ‘Good’ Sensor Orientation

As discussed in Section 3, all accelerations applied to the sensor are coplanar, lying within the rotation plane (Figure 2). Therefore, it is crucial to avoid orienting the sensor such that one of its sensing axes is orthogonal to this plane (i.e., parallel to the rotation axis), as this would result in no acceleration being detected, rendering calibration impossible. To ensure reliable calibration, a ‘good’ sensor orientation is one where all sensing axes are sufficiently distant from being parallel to the rotation axis, ensuring a substantial acceleration signal on each axis. Figure 5 illustrates this orientation, in which the sensor axes are tilted to align with the faces of a cube whose diagonal coincides with the radial axis. This configuration can be practically achieved by mounting the sensor on a precisely tilted support. We designed and fabricated such a support using 3D printing.

It is worth noting that the accuracy of the sensor positioning is not critical: the orientation can be fairly approximate, as long as it ensures that the sensing axes are not parallel to the rotation axis. The precise orientation is estimated by the calibration algorithm, which represents it using the misalignment matrix R.

### 3.3. The Accelerometer Model

We adopt the classical six-parameter linear model [4,13,16], which considers independent offset and gain components for each sensing axis. This model can be expressed as(6)$$v\left(t\right)=\left[\;\begin{array}{c} v_{x}\\ v_{y}\\ v_{z} \end{array}\;\right]=\left[\;\begin{array}{c} o_{x}\\ o_{y}\\ o_{z} \end{array}\;\right]+\left[\;\begin{array}{ccc} \gamma_{x} & 0 & 0\\ 0 & \gamma_{y} & 0\\ 0 & 0 & \gamma_{z} \end{array}\;\right]s\left(t\right)\;=\;o+\Gamma \;s\left(t\right)$$
where

v(t) is the output acceleration, as generated by the sensor under test.s(t) is the sensed acceleration, that is, the actual acceleration applied along the sensing axes $\langle xyz\rangle$.$o={\left[o_{x},o_{y},o_{z}\right]}^{T}$ and $\Gamma =\mathrm{diag}\left[\gamma_{x},\gamma_{y},\gamma_{z}\right]$ are the accelerometer offset and gain.

Combining Equations (Equation 1), (Equation 4), and (Equation 6), we obtain the complete calibration model, which relates the sensor output v(t) to the bench acceleration a(t):(7)v(t)=o+Γs(t)=o+ΓRa(t)
where offset $o={\left[o_{x},o_{y},o_{z}\right]}^{T}$ and gain $\Gamma =\mathrm{diag}\left[\gamma_{x},\gamma_{y},\gamma_{z}\right]$ are the sensor parameters: the actual target of the calibration.

### 3.4. The Calibration Problem

Referring to (Equation 7), R, o, and Γ are defined by three parameters each, and a(t) is described by the following six parameters: $\langle r,\vartheta_{0},\omega_{0}\ldots \omega_{3}\rangle$. Therefore, the complete calibration model is defined by a total of 15 parameters. We collect these into a 15-dimensional model vector m, defined as follows:(8)$$m=\left\{o,\;\Gamma ,\;R,\;r,\;\vartheta_{0},\;\omega_{0},\;\omega_{1},\;\omega_{2},\;\omega_{3}\right\}.$$

The solution of the calibration problem then consists in the determination of the model vector m that best fits the observed data. The fitting is evaluated as the least squares difference between the observed data $v_{obs}\left(t\right)$ and the predicted data $v_{pred}\left(m,t\right)$, generated using m in the model (Equation 7). The solution $\hat{m}$ can be expressed as follows:(9)$$\hat{m}=\underset{m}{\arg\min}\;{∥v_{obs}\left(t\right)-v_{pred}\left(m,t\right)∥}^{2}.$$

#### Problem Determinability

In [16], a similar setup was proposed, where the rotating bench was tilted at an unknown angle, rather than being vertical. In that case, it was demonstrated that the calibration problem becomes underdetermined, due to the presence of a one-dimensional null space within the parameter subspace $\langle r,\delta ,\Gamma ,o_{z}\rangle$, where δ represents the unknown tilt angle. This ambiguity can be resolved by fixing either *r* or δ, thereby removing the extra degree of freedom, as demonstrated in [16]. In contrast, the vertical bench setup used here makes the tilt angle δ known a priori (δ=π/2). As a result, the calibration problem defined in (Equation 9) is determined.

## 4. The Calibration Algorithm

The proposed calibration algorithm is an evolution of the technique proposed in [16]. The computational process consists of four main steps, as represented in the flow chart in Figure 6.

In the first step, the acquired acceleration v(t), referred to the sensor frame 〈xyz〉, is rotated in such a way as to align it, as much as possible, to the bench frame 〈XYZ〉. This alignment can be only partial because different values of gain along the sensing axes cause the relationship between v(t) and a(t), expressed in (Equation 7), to deviate from a simple 3D rotation. The details are described in Section 4.1.

The second step estimates the *motion parameters* $\langle \vartheta_{0},\omega_{0}\ldots \omega_{3}\rangle$ that best model the bench rotation ϑ(t) according to (Equation 3). As Section 4.2 shows, it is possible to estimate ϑ(t) by analyzing the sinusoidal components in the partially realigned signal obtained at the previous step.

The third step uses the previously estimated rotation phase ϑ(t) to determine the sensor’s misalignment matrix R and the gain Γ.

Finally, the fourth step estimates the remaining unknown parameter, the sensor offset o, as the constant additive term that best fits the acquired data v(t).

Each of the four steps are described in detail in the following subsections.

### 4.1. Partial Alignment

Referring to Figure 4, this step aims to perform the *partial alignment* of the acquired acceleration v(t), measured along the 〈xyz〉 axes, with the bench reference frame 〈XYZ〉. This alignment is performed by applying the *alignment rotation matrix*T to the data v(t), resulting in the rotated output r(t)=Tv(t).

According to (Equation 7), the application of a rotation T to the acquired data v(t) can be expressed as follows:(10)r(t)=Tv(t)=To+TΓRa(t)=To+Wa(t)
with W=TΓR, where Γ is the diagonal gain matrix, whereas T and R are rotation matrices.

As demonstrated in [16], if the gain were the same along all axes, that is, $\Gamma =\gamma \;I_{3}$, then W=γTR; therefore it could be possible to perfectly re-align the sensor frame 〈xyz〉 to the reference frame 〈XYZ〉 just by rotation, by making the alignment matrix T equal to the inverse of the misalignment matrix R (which, for rotation matrices, coincides with the transpose, $T=R^{T}$), thus leading to $W=\gamma \;I_{3}$ in (Equation 10).

However, in real triaxial accelerometers, the gains along different axes are slightly unequal. As a result, $\Gamma \neq \gamma \;I_{3}$, and consequently W=TΓR is no longer orthogonal, nor a rotation matrix.

Equation (Equation 10) can be seen, in its generality, as a change of reference of the 3-D vector a(t) from its Euclidean reference 〈XYZ〉 to a non-orthogonal reference $\langle x^{\prime }y^{\prime }z^{\prime }\rangle$:(11)$$r\left(t\right)=\left[\;\begin{array}{c} r_{x^{\prime }}\left(t\right)\\ r_{y^{\prime }}\left(t\right)\\ r_{z^{\prime }}\left(t\right) \end{array}\;\right]=T\;o+W\;a\left(t\right)=T\;o+\left[\;\begin{array}{ccc} w_{11} & w_{12} & w_{13}\\ w_{21} & w_{22} & w_{23}\\ w_{31} & w_{32} & w_{33} \end{array}\;\right]\;\left[\;\begin{array}{c} a_{X}\left(t\right)\\ a_{Y}\left(t\right)\\ a_{Z}\left(t\right) \end{array}\;\right]$$

If W is not orthogonal, $\langle x^{\prime }y^{\prime }z^{\prime }\rangle$ results in a non-orthogonal frame; therefore it is impossible to overlay $\langle x^{\prime }y^{\prime }z^{\prime }\rangle$ onto the Euclidean frame 〈XYZ〉 just by rotation.

This situation is illustrated in Figure 7. To align the non-orthogonal frame $\langle x^{\prime }y^{\prime }z^{\prime }\rangle$ as closely as possible with the Euclidean frame 〈XYZ〉, the following approach may be adopted:Apply first a rotation around *X* and *Y* that aligns $z^{\prime }$ to *Z*, thus making $z^{\prime }⊥X$ and $z^{\prime }⊥Y$.Apply then a rotation around $z^{\prime }\equiv Z$ that brings the axis $y^{\prime }$ into the YZ plane, thus making $y^{\prime }⊥X$.

If $z^{\prime }⊥X$, $z^{\prime }⊥Y$, and $y^{\prime }⊥X$, then $w_{21}=w_{31}=w_{32}=0$, which means that W turns out to be upper-triangular.

The alignment matrix T can be then expressed as the composition of these two rotations. Describing them with the rotation matrices $T^{\prime }$ and $T^{\prime \prime }$, respectively, we have $T=T^{\prime \prime }\;T^{\prime }$.

#### Estimation of $T^{\prime }$ and $T^{\prime \prime }$

The estimation of $T^{\prime }$ and $T^{\prime \prime }$ is performed as proposed in [16]. $T^{\prime }$ is the rotation of the frame $\langle x^{\prime }y^{\prime }z^{\prime }\rangle$ around the axes *X* and *Y* that brings $z^{\prime }$ to coincide with *Z*. Applying this rotation to v(t) yields the rotated data $r^{\prime }\left(t\right)=T^{\prime }\;o+W^{\prime }\;a\left(t\right)$, where $W^{\prime }=T^{\prime }\;\Gamma R$, and whose $z^{\prime }$ component results in the following:(12)$$r_{z^{\prime }}^{\prime }\left(t\right)=t_{3}^{\prime }o+w_{31}^{\prime }\;a_{X}\left(t\right)+w_{32}^{\prime }\;a_{Y}\left(t\right)$$

After rotation, $z^{\prime }$ turns out to be orthogonal to both *X* and *Y*, therefore $w_{31}^{\prime }=w_{32}^{\prime }=0$. Thus, $r_{z^{\prime }}^{\prime }\left(t\right)$ reduces to a constant: $r_{z^{\prime }}^{\prime }\left(t\right)=t_{3}^{\prime }o$.

Considering that real data v(t) are noisy, the condition of constant $r_{z^{\prime }}^{\prime }\left(t\right)$ corresponds to the condition of its minimum variance. The estimation of $T^{\prime }$ is therefore performed by searching for the minimum variance of $r_{z^{\prime }}^{\prime }\left(t\right)$ in the 2D space of the two rotations around the *X* and *Y* axes, $\langle \varepsilon_{x},\varepsilon_{y}\rangle$:(13)$$T^{\prime }\left({\hat{\varepsilon }}_{x},{\hat{\varepsilon }}_{y}\right)=\underset{\varepsilon_{x},\;\varepsilon_{y}}{\arg\min}\left\{\mathrm{Var}\left(r_{z^{\prime }}^{\prime }\left(t\right)\right)\right\}.$$

The second rotation $T^{\prime \prime }$ is applied around the *Z* axis, now coincident with $z^{\prime }$, to make the $y^{\prime }$ axis lie in the tangential plane YZ and therefore to make it orthogonal to the radial axis *X*. Considering now the fully rotated data r(t)=Tv(t) defined in (Equation 11), where $T=T^{\prime \prime }T^{\prime }$, let us focus on the $y^{\prime }$ component, $r_{y^{\prime }}\left(t\right)$:

The second rotation $T^{\prime \prime }$ is applied around the *Z* axis (which now coincides with $z^{\prime }$) in order to align the $y^{\prime }$ axis with the tangential plane YZ, making it orthogonal to the radial axis *X*. Now, considering the fully rotated data r(t)=Tv(t), as defined in (Equation 11) with $T=T^{\prime \prime }T^{\prime }$, we focus on the $y^{\prime }$ component, $r_{y^{\prime }}\left(t\right)$:(14)$$r_{y^{\prime }}\left(t\right)=t_{2}o+w_{21}\;a_{X}\left(t\right)+w_{22}\;a_{Y}\left(t\right)$$
where $t_{2}$ is the second row of T.

As demonstrated in [16], $T^{\prime \prime }$ can be determined as the rotation around *Z* that minimizes the *excursion* of $r_{y^{\prime }}\left(t\right)$, defined as $\max\left[r_{y^{\prime }}\left(t\right)\right]-\min\left[r_{y^{\prime }}\left(t\right)\right]$. We can solve this problem as a minimum search in the 1D space of the rotation angle ${\hat{\varepsilon }}_{z}$ that defines $T^{\prime \prime }$:(15)$$T^{\prime \prime }\left({\hat{\varepsilon }}_{z}\right)=\underset{\varepsilon_{z}}{\arg\min}\;\left\{\max\left[r_{y^{\prime }}\left(t\right)\right]-\min\left[r_{y^{\prime }}\left(t\right)\right]\right\}.$$

### 4.2. Estimation of the Angular Velocity

After the previous data alignment, $w_{21}=0$ in (Equation 14); therefore $r_{y^{\prime }}\left(t\right)$ is as follows:(16)$$r_{y^{\prime }}\left(t\right)=t_{2}o+w_{22}\;a_{Y}\left(t\right)\approx t_{2}o+w_{22}\;g\sin\vartheta \left(t\right),$$
that is, a constant term plus a sinusoidal component with time-varying frequency, which can be modeled as(17)$$r_{y^{\prime }}\left(t\right)=k+A\sin\vartheta \left(t\right)\;:\;\;\left\{\;\begin{array}{c} A=w_{22}\;g\\ k\;=t_{2}o \end{array}\right.$$
with the phase ϑ(t) modeled by the motion parameters $\langle \vartheta_{0},\omega_{0}\ldots \omega_{3}\rangle$ in (Equation 3) ( as shown in [16], the tangential acceleration term $r\ddot{\vartheta }\left(t\right)$ in the acceleration model (Equation 1) can be neglected at this stage due to its negligible magnitude; the complete model will be considered in subsequent steps for the final estimations).

It is therefore possible to determine the motion parameters, along with *A* and *k*, as the values that, applied in (Equation 3) and (Equation 17), yield the best approximation of $r_{y^{\prime }}\left(t\right)$. The estimation can be formalized as a least-squares minimum search in the seven-dimensional parameter space $\langle \vartheta_{0},\omega_{0}\ldots \omega_{3},A,k\rangle$ as follows:(18)$$\langle \;{\hat{\vartheta }}_{0},{\hat{\omega }}_{0}\ldots {\hat{\omega }}_{3},\hat{A},\hat{k}\;\rangle =\underset{\vartheta_{0},\omega_{0}\ldots \omega_{3},A,k\;}{\arg\min}{∥\;r_{y^{\prime }}\left(t\right)-\left[k+A\sin\vartheta (t)\right]\;∥}^{2}.$$

### 4.3. Estimating Misalignment, Gain, and Radius

Once the bench rotation ϑ(t) is known, the next step involves jointly estimating the sensor misalignment matrix R, the gain matrix Γ, and the rotation radius *r*.

For this step, we leverage the constraint that W is upper triangular. Specifically, we focus on the following coefficients of W:(19)$$\left\{\begin{array}{c} w_{21}=t_{21}\gamma_{x}r_{11}+t_{22}\gamma_{y}r_{21}+t_{23}\gamma_{z}r_{31}=0\\ w_{31}=t_{31}\gamma_{x}r_{11}+t_{32}\gamma_{y}r_{21}+t_{33}\gamma_{z}r_{31}=0\\ w_{32}=t_{31}\gamma_{x}r_{12}+t_{32}\gamma_{y}r_{22}+t_{33}\gamma_{z}r_{32}=0\\ w_{22}=t_{21}\gamma_{x}r_{12}+t_{22}\gamma_{y}r_{22}+t_{23}\gamma_{z}r_{32}=\frac{g}{A} \end{array}\right.$$
where $t_{ij}$ are the known coefficients of T; $r_{ij}$ are the coefficients of R, which are functions of the unknown Euler angles 〈θ,ϕ,ψ〉; the gain vector $\gamma =\langle \gamma_{x},\gamma_{y},\gamma_{z}\rangle =\mathrm{diag}\left(\Gamma \right)$ contains the three gain components. The term g/A is taken as the estimation of $w_{22}$, according to (Equation 17), with *A* previously estimated solving (Equation 18).

Assuming that R is known, the four equations in (Equation 19) define a non-homogeneous, overdetermined linear system with three unknown gain components $\langle \gamma_{x},\gamma_{y},\gamma_{z}\rangle$. Therefore, for any given value of R, the system (Equation 19) can be used to compute the corresponding gain vector $\gamma =\langle \gamma_{x},\gamma_{y},\gamma_{z}\rangle$. Since the system is overdetermined, it is solved in the least-squares sense using the pseudo-inverse method [19]. By expressing (Equation 19) in matrix form, we define the coefficient matrix as U, and the right-hand side as $b={\left[\;0\;0\;0\;g/A\;\right]}^{T}$. The overdetermined system is then given by Uγ=b. The best solution for γ in the least-squares sense is therefore(20)$$\gamma ={\left(U^{T}U\right)}^{-1}U\;b.$$

The combined estimation of R, *r*, and Γ can be therefore formulated as a nonlinear optimization problem over the four-dimensional space 〈r,θ,ϕ,ψ〉 of the rotation radius *r* and the three Euler angles defining R. For each candidate rotation matrix R(θ,ϕ,ψ) selected by the search algorithm, the corresponding gain vector γ is computed using (Equation 20). The objective function is defined as the norm of the difference between the realigned acquired data, r(t)=Tv(t), and the modeled data, $r^{m}\left(t\right)=Tv^{m}\left(t\right)$, where $v^{m}\left(t\right)$ depends on the parameters 〈r,θ,ϕ,ψ〉 and is evaluated according to the complete model in (Equation 7). Prior to computing the norm, both signals are mean-subtracted to remove any DC bias:(21)$$\begin{array}{c} r_{0}\left(t\right)=r\left(t\right)-\mathrm{mean}\left[r(t)\right]\\ r_{0}^{m}\left(t\right)=r^{m}\left(t\right)-\mathrm{mean}\left[r^{m}\left(t\right)\right]; \end{array}$$
the minimization problem is therefore defined as follows:(22)$$\hat{R}=\underset{R}{\arg\min}{∥r_{0}\left(t\right)-r_{0}^{m}\left(t\right)∥}^{2}.$$

The use of the zero-mean components $r_{0}\left(t\right)$ and $r_{0}^{m}\left(t\right)$ in (Equation 22) is essential to ensure that the objective function is independent of the offset o, which remains unknown. This decouples the estimation of R and Γ from that of o, thereby simplifying the optimization process by reducing the dimensionality of the search space.

To obtain $r^{m}\left(t\right)$, the modeled signal $v^{m}\left(t\right)$ is computed using the full model in (Equation 7), with an arbitrary value assigned to o. This choice is justified, as $r_{0}^{m}\left(t\right)$ is subsequently forced to have zero mean, rendering the specific value of o irrelevant. Once $v^{m}\left(t\right)$ is computed, the modeled realigned data are obtained as $r^{m}\left(t\right)=T\;v^{m}\left(t\right)$. For this estimation and those that follow, the complete model (Equation 1) for a(t) is considered, with the tangential acceleration component $r\ddot{\vartheta }\left(t\right)$ no longer neglected.

### 4.4. Estimation of the Offset

The final unknown parameter, the offset o, can be estimated by performing a minimization in the three-dimensional space defined by its components $\langle o_{x},o_{y},o_{z}\rangle$. This involves minimizing the norm of the difference between the originally acquired data v(t) and the synthesized data $v^{m}\left(t\right)$, computed using (Equation 7) with the previously determined parameters. The corresponding minimization problem can be formulated as follows:(23)$$\hat{o}=\arg\min_{o}{∥\;v\left(t\right)-v^{m}\left(t\right)\;∥}^{2}.$$

All optimization problems in the proposed procedure are solved using a nonlinear minimization algorithm, specifically the Nelder–Mead algorithm [20]. In all experiments performed, this algorithm never exhibited stability or convergence issues, likely due to the absence of local minima in the defined search spaces, thereby demonstrating its effectiveness as an optimization strategy for this application.

## 5. Results

To assess the performance of the proposed procedure, I implemented the two tests outlined in a similar approach [16]. The first test involves running the calibration on synthetically generated data, for which the exact values of the calibrated parameters are known a priori. This test aims to evaluate the accuracy of the estimated parameters as a function of the noise added to the data. In the second test, the calibration is evaluated experimentally by calibrating a commercial triaxial MEMS accelerometer. This experiment aims to measure the actual precision of the calibrated parameters, expressed in terms of the standard deviation of multiple estimations obtained from repeated calibrations of the same sensor [21].

### 5.1. Measuring Accuracy and Precision with Synthetic Data

In this test, the calibration is performed on several realizations of synthetic data generated by applying the model Equations (Equation 7) to a set of arbitrarily chosen parameters representing the true values. Each realization is supplemented with a different instance of zero-mean Gaussian noise, simulating the noise that affects the ideal acceleration data in real accelerometers. The aim of this test is to assess the robustness and accuracy of the algorithm as a function of the noise level affecting the ideal data. The accuracy and precision of the estimations are derived from the means and variances of the calibration results.

To reflect the noise characteristics commonly found in commercial-grade MEMS accelerometers [18], I selected standard deviation values logarithmically spaced between 0.01 m/s^2^ and 1 m/s^2^. For each selected noise level, I conducted 60 independent calibration trials with different random noise instances to obtain statistically meaningful results. Synthetic data were generated to mimic real measurements: for all calibrated parameters, I selected typical values observed in actual calibration procedures. For each calibration experiment, a synthetic 90 s sequence was produced at a sampling rate of 200 samples per second, exactly as in the real calibrations (Section 5.2).

The mean values of each parameter serve as a good indicator of the method’s accuracy, revealing any biases in the estimation. The standard deviations, on the other hand, provide an estimate of the achievable precision.

The mean values are plotted in Figure 8a, as a function of the standard deviation of the Gaussian noise added to the data. For each parameter, the normalized deviation $\mu_{dev}$ is plotted, defined as the difference between the mean value $\mu_{cal}$ and true values $v_{true}$, normalized by the true value itself and expressed as percentage: $\mu_{dev}=100\left(\mu_{cal}/v_{true}-1\right)$.

To ensure consistency in the representation, deviations of similar parameter groups were aggregated by computing their mean values. The plotted curves correspond to

Sensor offset (mean of *x*, *y*, and *z* components);Sensor gain (mean of *x*, *y*, and *z* components);Rotation radius;Misalignment (mean of the three Euler angles);Motion parameters (mean of the five components).

The results indicate that, for low noise levels, the average deviation $\mu_{dev}$ is approximately zero, implying that the estimated values $\mu_{cal}$ closely match the true parameters $v_{true}$. This supports the unbiased nature of the proposed estimation method. Furthermore, the estimation remains effectively unbiased (with $\left|\mu_{dev}\right|<0.25\%$) up to a noise level of $\sigma_{n}=$ 0.5 m/s^2^, well above the typical noise observed in commercial low-*g* MEMS accelerometers [17,18].

The obtained deviations from the mean are considerably lower than those obtained in [16]: in particular, regarding the actual calibration parameters, gain and offset, the reduction is substantial—the gain deviation decreases from approximately 2% to 0.6%, and the offset deviation decreases from 1.2% to 0.14% (considering the worst case for both). This improvement highlights the superior performance of the proposed approach over that in [16], particularly in terms of reduced estimation biases and, consequently, enhanced accuracy.

Figure 8b presents the standard deviations of the estimated parameters, normalized by their respective mean values ( for gain and offset, I used $\mu_{o}=\mu_{\gamma }=1$, so their normalized deviations in Figure 8b coincide numerically with the standard deviations), as a function of the data noise standard deviation. As in Figure 8a, similar variables are grouped by averaging to ensure consistent representation. From this diagram, the following conclusions can be drawn:The standard deviation of all parameters increases roughly proportionally to the data noise across the entire range, without signs of divergence. This behavior indicates that the estimation method remains robust even under noise levels well above those commonly encountered in MEMS accelerometers.The estimation of the rotation radius *r* exhibits the lowest standard deviation, supporting the validity of the choice to estimate the radius through calibration in the proposed setup.The precision in the estimation of the sensor’s gain is one order of magnitude higher than that of the sensor’s offset across the entire noise range.

The results shown in Figure 8a,b demonstrate that, despite the simplicity of the proposed setup, the calibration achieves high levels of accuracy and precision. For the typical noise levels of commercial-grade MEMS accelerometers [17,18], the precision achieved is highly satisfactory, further reinforcing the effectiveness of the proposed setup.

### 5.2. Performance Assessment on a Real Sensor

I adopted the same assessment approach as in [16]: I repeated the calibration procedure multiple times on the same sensor, with each iteration yielding a measurement of the sensor parameters. The mean value and standard deviation of the measurements for each parameter offer a reliable estimation of the achieved precision [21].

I constructed a simple calibration bench using a common bicycle wheel as a rotating base, as shown in Figure 1. A photograph of the calibration setup is shown in Figure 9. I designed a wireless accelerometer system in which the sensor under calibration—a digital MEMS accelerometer (Invensense™ MPU9250 [18])—is interfaced with a battery-powered microcontroller board (ESP32-Wroom [22]). The board features a Bluetooth interface used for both wireless data acquisition and device control.

The rotating bench was firmly anchored to the floor, and its plane was aligned vertically using a level, as described in Section 2.2. The sensing device was installed on the calibration bench together with a counterbalance, whose position was adjusted to achieve equilibrium of the wheel. To achieve the sensor orientation described in Section 3.2 and shown in Figure 5, a custom support was designed and 3D-printed (its shape is depicted in Figure 10). This support rigidly connects the sensor to the MCU board and maintains the desired tilt. For data acquisition, the wheel was manually spun, and data were collected from the sensor at a sampling rate of 200 Hz. A 90 s sequence (approximately corresponding to 100 wheel rotations) was extracted and used for calibration. This acquisition process was repeated 10 times with the same setup (i.e., the same sensor positioned identically on the bench), though each trial involved different motion due to the manual spinning of the bench. Each data acquisition was followed by a calibration procedure, resulting in an estimate of the sensor parameters. Table 1 reports the mean and standard deviation of each parameter across the 10 calibration runs.

### 5.3. Precision Assessment

Since the synthetic data exactly correspond to the model in (Equation 1), computed using known true values, and the added noise is Gaussian with zero mean, the standard deviations plotted in the diagram of Figure 8b can be reasonably considered an estimate of the precision achievable by this technique, as a function of the noise level in the data.

The level of noise affecting the real data has been estimated experimentally, exploiting the fact that the minimized variance of $r_{z^{\prime }}\left(t\right)$, after the minimization in (Equation 13) (described in Section 4.1), is a reliable estimate of the noise variance: without noise, the minimization would lead to a constant $r_{z^{\prime }}\left(t\right)$, so its minimum variance would be zero. Consequently, the minimum reached by (Equation 13) can be reasonably taken as an estimation of the noise variance on the acquired data.

By averaging the results obtained from all calibrations, I estimated a noise standard deviation of approximately $\sigma_{n}\approx$ 0.52 m/s^2^. For this noise level, the plots in Figure 8b indicate the following expected standard deviations for gain and offset:(24)$$\sigma_{\mathrm{gain}}^{\exp}\approx 2.5\times 10^{-3},\;\;\;\;\;\sigma_{\mathrm{offset}}^{\exp}\approx 4\times 10^{-2}\;{m/s}^{2}.$$

In comparison, the measured standard deviations of the calibrated parameters, computed by averaging the *x*, *y*, and *z* components of both gain and offset reported in Table 1, are(25)$$\sigma_{\mathrm{gain}}^{\mathrm{meas}}=4.1\times 10^{-3},\;\;\;\;\;\sigma_{\mathrm{offset}}^{\mathrm{meas}}=3.9\times 10^{-2}\;{m/s}^{2}.$$

The measured standard deviations closely match the expected values, demonstrating strong agreement between the proposed model in (Equation 7) and the actual calibration setup. This supports the reliability of the calibration results produced by the method.

#### Evaluation of the Noise Sources

The measured standard deviation $\sigma_{n}$ refers to the total noise affecting the data, which arises from multiple contributing phenomena. These contributions can be categorized into two main noise components:The *sensor noise*, including all the noise sources originated by the sensor itself.The *bench noise*, including the noise sources due to all mechanical non-idealities (such as friction irregularities and clearances in the ball bearings, vibration of the supports, etc.) that give rise to undesired and unpredictable deviations from the ideal acceleration defined by the dynamical model (Equation 1).

Assuming that these two noise components are zero-mean and statistically independent, the total noise variance (i.e., total noise power) $\sigma_{n}^{2}$ is given by the sum of their individual variances:(26)$$\sigma_{n}^{2}=\sigma_{S}^{2}+\sigma_{B}^{2}$$
where $\sigma_{S}^{2}$ and $\sigma_{B}^{2}$ denote the variances of the sensor noise and bench noise, respectively.

Since it was possible to measure the standard deviation of the total noise $\sigma_{n}$, and the standard deviation of the sensor noise $\sigma_{S}$ is normally provided by the manufacturer, we can derive the level of the bench noise $\sigma_{B}$, due to all non-idealities of the calibration bench. For the sensor used in this experiment [18], the datasheet reports $\sigma_{S}\approx 0.08$ m/s^2^. Thus, according to (Equation 26), we obtain the estimate for the standard deviation of the bench noise: $\sigma_{B}=\sqrt{\sigma_{n}^{2}-\sigma_{S}^{2}}\approx 0.51$ m/s^2^. This result indicates that, for the accelerometer tested, the overall data noise is largely dominated by the bench noise.

When considered alongside the experimental accuracy results reported in Table 1 and the theoretical accuracy results in Figure 8, this finding supports the following conclusions:The proposed method is capable of achieving accurate parameter estimation even under relatively noisy conditions.Nonetheless, improving the mechanical quality of the setup—while maintaining its low-cost nature—could further reduce the bench noise and thereby enhance significantly the calibration accuracy.

## 6. Conclusions

The paper presents a dynamic calibration method for triaxial accelerometers working with a calibration setup, which is very simple and uncritical to build. The experimental calibration tests have shown, primarily, that this method yields noticeable accuracy and precision, despite the extreme simplicity of the setup. This fact makes the proposed method particularly suitable for low-cost calibrations of commercial-grade accelerometers.

As outlined in the introduction and supported by the results, the proposed method builds upon the approach presented in [16]. In addition to eliminating the underdetermined nature of the calibration problem and addressing the issue of suboptimal sensor orientations (discussed in Section 3.2), it also results in a significant improvement in achievable accuracy. This enhancement is demonstrated by the experimental results with synthetic data, presented in Section 5.1.

Moerover, the analysis in Section 5.3 has shown that an improvement in the build quality of the bench could further enhance the precision of this calibration.

It is also worth noting that the analysis in Section 5.1 shows that the accuracy of this method scales proportionally with the noise level in the data. This makes this dynamic calibration approach potentially applicable to higher-precision accelerometers, through the use of more advanced sensor models that account for effects not captured by static calibration methods, such as gain nonlinearities, non-orthogonality of the sensing axes, and hysteresis.

## Figures and Tables

**Figure 1 sensors-25-03998-f001:**
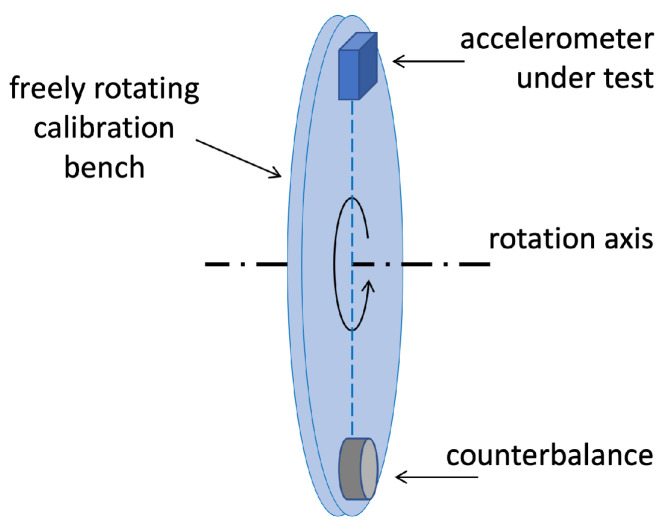
The calibration setup. The accelerometer for calibration is mounted on a rotating bench, together with a properly placed counterbalance, to balance the rotation. The bench plane is adjusted to be vertical.

**Figure 2 sensors-25-03998-f002:**
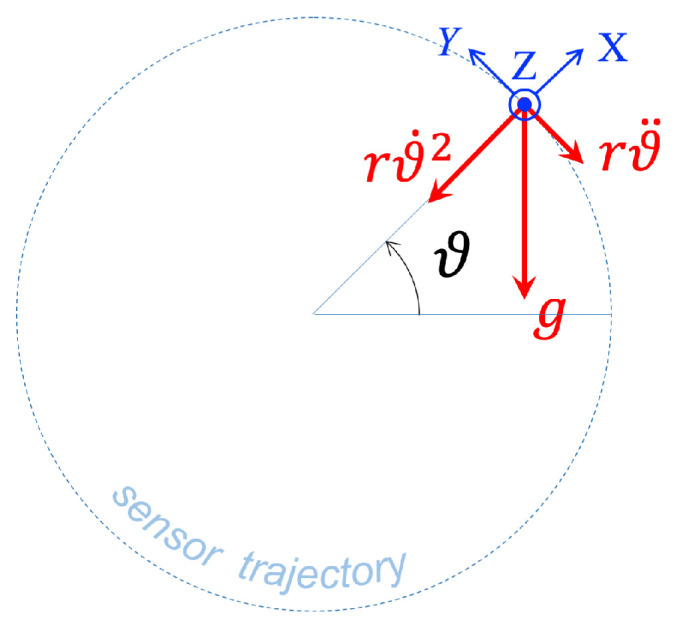
Geometric description of the bench motion. While rotating, the sensor undergoes three accelerations: the radial centripetal acceleration ${\dot{\vartheta }}^{2}r$, gravity *g* along the vertical, and the tangential deceleration $r\ddot{\vartheta }$, caused by friction.

**Figure 3 sensors-25-03998-f003:**
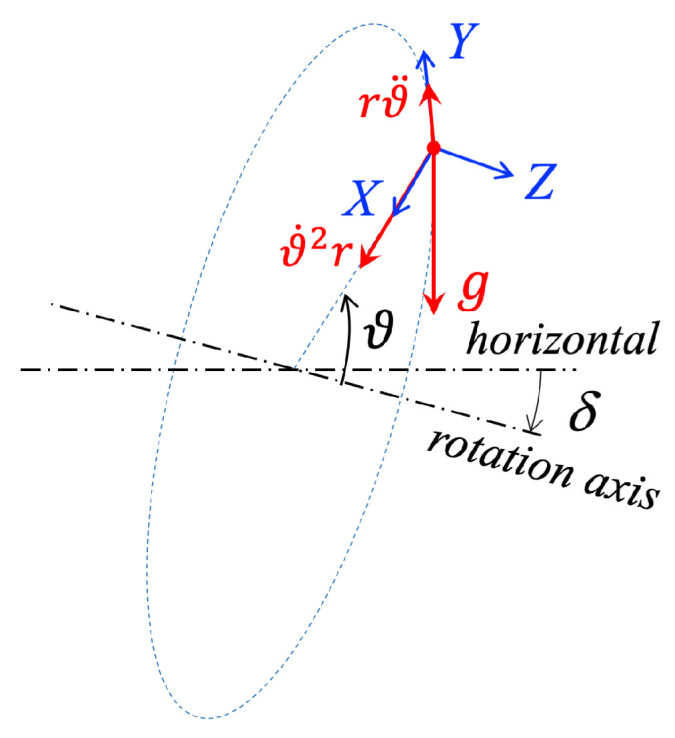
Geometric representation of a ‘nearly vertical’ bench, with the rotation axis tilted by δ. In this case, since *g* is vertical, the three accelerations are no longer coplanar.

**Figure 4 sensors-25-03998-f004:**
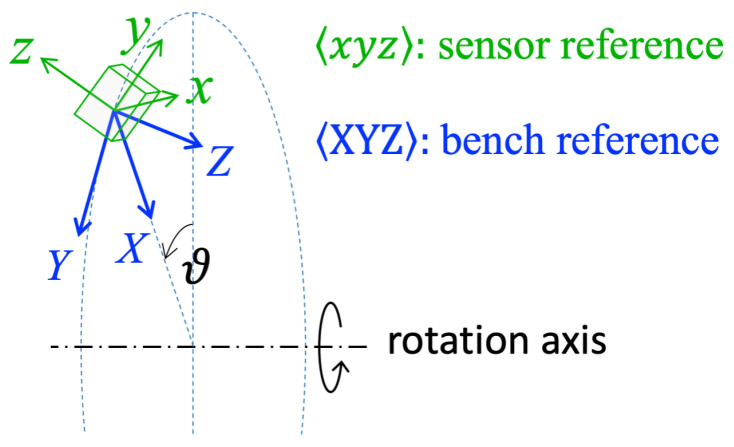
Schematization of the sensor misalignment. The actual orientation of the sensing reference frame is unknown. This misalignment can be modeled as the 3D rotation that overlaps the bench reference frame 〈XYZ〉 onto the sensor reference frame 〈xyz〉.

**Figure 5 sensors-25-03998-f005:**
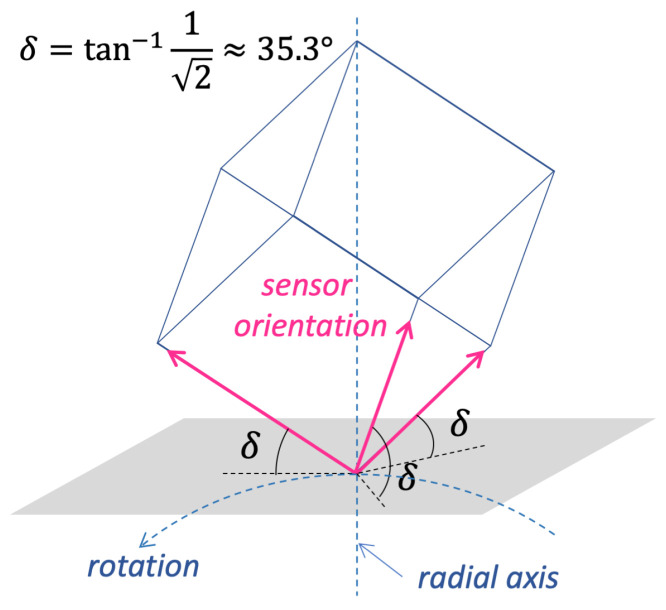
Best sensor orientation. By placing the sensor with the three axes oriented as the sides of a tilted cube having its diagonal along the radial axis, all the sensor axes are tilted on the same angle δ with respect to the tangential plane.

**Figure 6 sensors-25-03998-f006:**
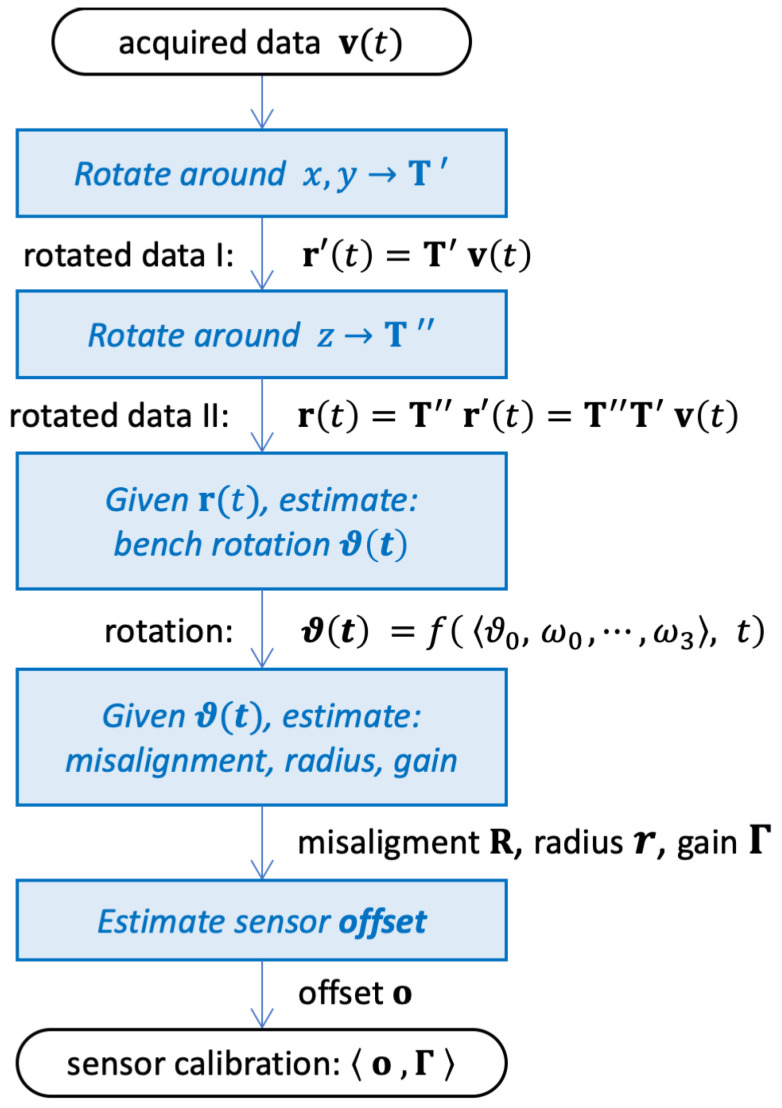
Flowchart of the proposed calibration. The blocks correspond to the computation steps described in Section 4.1, Section 4.2, Section 4.3 and Section 4.4.

**Figure 7 sensors-25-03998-f007:**
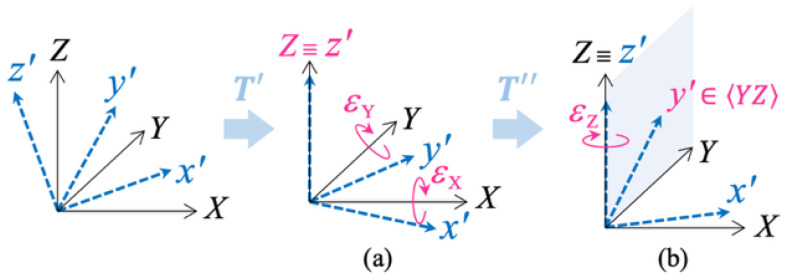
Schematization of the partial alignment step. The reference frame $\langle x^{\prime }y^{\prime }z^{\prime }\rangle$ is not orthogonal; therefore it cannot be aligned to 〈XYZ〉 by rotation. However, $\langle x^{\prime }y^{\prime }z^{\prime }\rangle$ can be rotated so that (**a**) $z^{\prime }\equiv Z$ (rotation matrix: $T^{\prime }$) and then (**b**) $y^{\prime }\in \langle YZ\rangle$ (rotation matrix: $T^{\prime \prime }$). The alignment matrix T corresponds to the composition of the following two rotations: $T=T^{\prime \prime }T^{\prime }$.

**Figure 8 sensors-25-03998-f008:**
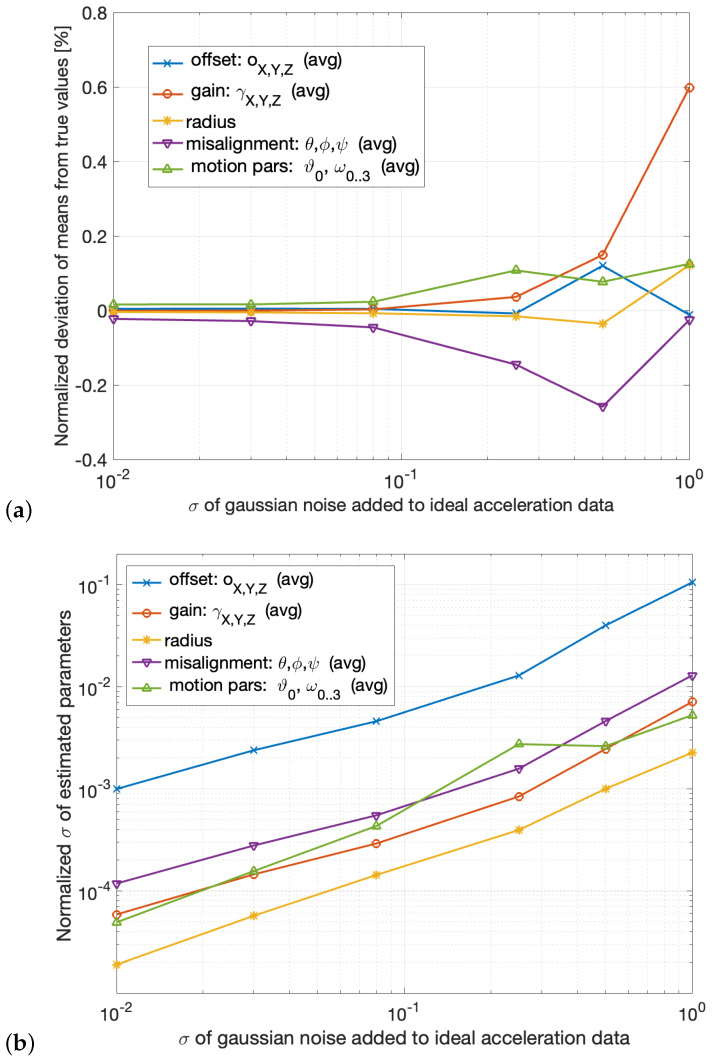
(**a**) Mean value of the estimated parameters, expressed as normalized difference from the true value. (**b**) Standard deviation of the estimated parameters, normalized to their mean value (σ/μ), for different levels (standard deviation) of Gaussian noise added to the ideal acceleration data.

**Figure 9 sensors-25-03998-f009:**
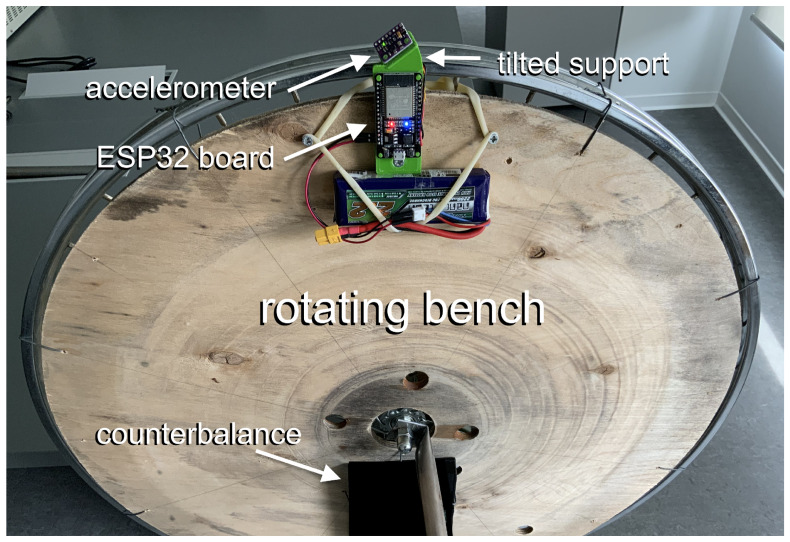
Prototype of the calibration setup, assembled by the author. The tilted support, used to give the sensing axes approximately the orientation shown in Figure 5, was custom-designed and 3D printed for this experiment. The sensor is rigidly connected to the MCU board via this support.

**Figure 10 sensors-25-03998-f010:**
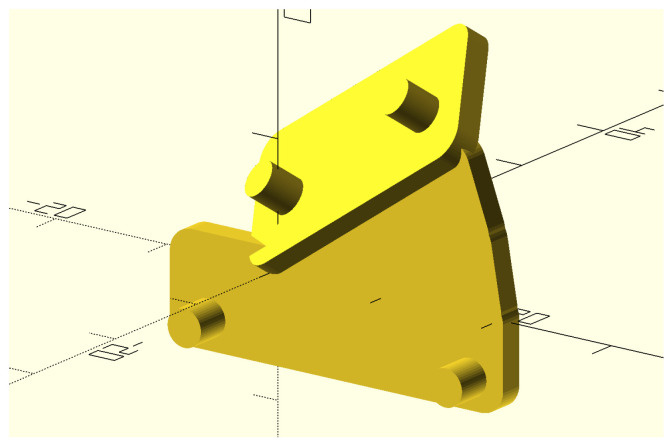
The shape of the 3D-printed support fastening the accelerometer under test to the MCU board and holding the sensor in the tilted orientation discussed in Section 3.2 and shown in Figure 5.

**Table 1 sensors-25-03998-t001:** Results of repeated calibrations of MEMS accelerometer: mean and standard deviation of calibrated parameters.

Parameter		Mean	Std. Deviation
Radius [mm]	*r*	328.12	0.192
Misalignment	θ	$\;\;7.55^{\circ }$	$0.152^{\circ }$
	ϕ	$44.74^{\circ }$	$0.356^{\circ }$
	ψ	$25.44^{\circ }$	$0.674^{\circ }$
Gain	$\gamma_{x}$	1.0553	0.00631
	$\gamma_{y}$	0.9996	0.00030
	$\gamma_{z}$	0.9580	0.00571
Offset [m/s^2^]	$o_{x}$	−0.182	0.0223
	$o_{y}$	+0.102	0.0640
	$o_{z}$	+0.027	0.0319

## Data Availability

Any inquiries regarding the data presented in this article should be directed to the corresponding author.

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
