# Peer review of "Simple Calibration of Three-Axis Accelerometers with Freely Rotating Vertical Bench"

_sensors, 2025, doi:10.3390/s25133998_

Round 1

Reviewer 1 Report

Comments and Suggestions for Authors

This paper proposes a dynamic calibration method for a three-axis accelerometer based on a vertical rotation table. The innovation lies in achieving high-precision calibration using a simple device and reducing the reliance on prior knowledge. Through theoretical modeling, algorithm design and experimental verification, the effectiveness of this method on synthetic data and real MEMS accelerometers has been proved. Experiments show that this method can achieve unbiased estimation in a low-noise environment, and its accuracy is superior to that of existing methods.However, some theoretical and experimental details could be further clarified.  Some contents worth modifying are as follows:

  1. Insufficient comparison of research status: This paper cites multiple literatures (such as [14]) to compare static and dynamic calibration methods, but lacks discussion on the new progress in low-cost MEMS calibration in recent years (after 2020).
    1. Data source annotation: Experimental data (such as the noise level of MPU9250) should clearly indicate whether they are from the manufacturer's manual (such as [16]) or independent test results.
    2. Picture 9 should be indicated whether it was built by the author themselves or referenced from third-party equipment (such as the design source of the 3D printing support structure). It is suggested to add dimension annotations (such as tilt angles) to Picture 10 and indicate whether the design file is open source.
    3. The original text only describes that the device is composed of "freely rotating wheels and counterweights", but does not explain its core physical principles (such as conservation of angular momentum and energy dissipation mechanism). The lack of theoretical support will reduce the scientific nature of the method.
    4. The original text uses a cubic polynomial model (Equation 3) to describe angular velocity decay, but fails to explain its correlation with real physical processes (such as frictional damping and air resistance), which may cause readers to question the arbitrariness of the model.
    5. In The "The Calibration Algorithm" section of the article, when there are significant differences in sensor gain on different axes, how to ensure the accurate estimation of the rotation matrix T, and how this method performs in the face of actual sensor noise and errors, insufficient experimental data or theoretical analysis are not provided in the article.
    6. In The "Calibration Algorithm" section of the article, in the steps of estimating the Misalignment and Gain of the sensor, the author adopted the method of nonlinear optimization search. However, nonlinear optimization search usually faces problems of high computational complexity and uncertain convergence. The specific implementation of the selected nonlinear optimization algorithm, the initial value selection strategy and the convergence proof are not discussed in detail in the text.
    7. In the noise source analysis of the Results section, the author distinguished between sensor noise and calibration table noise and estimated their contributions to the total noise. However, this analysis is relatively superficial and does not delve deeply into the specific sources and characteristics of the noise. For instance, regarding the noise of the calibration table, the author only mentioned factors such as friction irregularity and bearing clearance, but did not further analyze how these factors specifically affect acceleration measurement or how to reduce this noise by improving the design of the calibration table.

Reviewer 2 Report

Comments and Suggestions for Authors

The paper considers a dynamic calibration procedure for triaxial accelerometers characterized by a very simple and low-cost setup, where the calibration bench consists of a vertically, freely rotating wheel. It develops the method previously proposed by the authors in Pedersini, F. Dynamic Calibration of Triaxial Accelerometers With Simple Setup. IEEE Sensors Journal 2022, 22, 9665–9674. https://doi.org/10.1109/JSEN.2022.3164362.] The feature of the technique proposed in this paper is the accurate alignment of the table in a vertical plane, which is carried out using a leveling device. The precise alignment  makes it possible to ensure the observability  of all parameters and increase the accuracy of the estimate, which is demonstrated by experimental data.

Notes

1. There is no complete overview of accelerometer calibration methods. It is necessary to expand the review and indicate the advantages of the proposed method in relation to existing methods, and not only in relation to the previous one developed by the authors.

For example, the authors need to add links to papers A and B. In A, the radii to the accelerometers on the stand are estimated. In B, a manual bench is proposed for calibrating MEMS sensors. But in both papers, data from accelerometers and gyroscopes are considered, here only accelerometers are experimental data.

A. Emel’yantsev, G.I., Blazhnov, B.A., Dranitsyna, E.V. et al. Calibration of a precision SINS IMU and construction of IMU-bound orthogonal frame. Gyroscopy Navig. 7, 205–213 (2016). https://doi.org/10.1134/S2075108716030044

B. Vyalkov, A.V., Vyalkova, T.P. Calibration of IMU MEMS Sensors with the Use of a Manual Calibration Test Rig. Gyroscopy Navig. 14, 113–128 (2023). https://doi.org/10.1134/S2075108723020086

2. When describing the Dynamic Calibration Mode, a number of assumptions are used. The authors assume that the plane of the bench is strictly orthogonal to the axis of rotation. This is difficult to ensure, given the design of the bench. Non-orthogonality will lead to the presence of harmonic components in the accelerometer signals at the speed of rotation of the bench, which can lead to a biased estimate of the calibrated parameters.

3. The misalignment matrix R includes the non-orthogonality of the axes of sensitivity of the accelerometers among themselves and their rotation relative to the landing plane. These values are not taken into account in the accelerometer error model. What is the level of these values and can they be ignored?

3. These values may be negligible for MEMS sensors, but the conclusion is that this technique can be applied to more accurate sensors. However, without taking into account the components described above, this technique is not applicable for precision sensors.

4. In equation (16), when determining the rotation parameters, there is no component caused by angular acceleration from equation (1).

5. In the Calibration Algorithm, there is no stage for determining the distance from the accelerometer to the axis of rotation r (rotation radius).

6. Table 1 does not specify estimates of the wheel rotation parameters. 

Reviewer 3 Report

Comments and Suggestions for Authors

The study is devoted to the problem of dynamic calibration of a triad of accelerometers on a stand with a priori unknown motion parameters. The author sought to simplify the stand and the calibration algorithm as much as possible. Therefore, the entire calibration is divided into several successive stages: determination of the alignment rotation matrix; evaluation of motion parameters; evaluation of the actual sensor misalignment matrix and the sensor gain. The proposed calibration of the triad of accelerometers is tested using simulation and experiments. The manuscript is well structured. The research is described so clearly that it can even be used to teach the basics of calibration.But the manuscript has several shortcomings that need to be corrected.

Comments:

1) The manuscript must provide several examples of specific tasks for which factory calibration (calibration performed by the accelerometer manufacturer) is insufficient, but the calibration proposed in the manuscript is sufficient.

2) In section 2.2 or section 3, you need to list what was neglected when compiling the dynamic calibration model: bearing beating, elastic vibrations of the stand disk, etc. And you need to note what this should presumably lead to (i.e., to an increase in “bench noise”). You also need to describe the minimum requirements for the non-parallelism of the measuring axes to the rotation axis Z. That is, describe not only the “Good” sensor orientation, but also the “minimum permissible” sensor orientation.

3) In section 4, you need to briefly describe how the stand moves during each stage of calibration: is it continuously rotated by hand or does it rotate freely during measurements (accelerated before measurements) or something else.

4) In section 5.1, you should describe the number of "measurements" (e.g. long sequence and sampling rate) for which the calibration simulation was performed.

5) It is necessary to explain the reason for the appearance of a local maximum of standard deviation of motion parameters.

6) In section 5, you need to add a comparison of the calibration results with the factory calibration results (if they exist for the sensors used in the experiment).

Round 2

Reviewer 1 Report

Comments and Suggestions for Authors

The authors have addressed all my comments.

Reviewer 2 Report

Comments and Suggestions for Authors

The comments have been corrected.